# Size Control of Biomimetic Curved-Edge Vaterite with Chiral Toroid Morphology via Sonochemical Synthesis

**DOI:** 10.3390/biomimetics9030174

**Published:** 2024-03-13

**Authors:** Ki Ha Min, Dong Hyun Kim, Seung Pil Pack

**Affiliations:** 1Institute of Industrial Technology, Korea University, Sejong 30019, Republic of Korea; alsrlgk@gmail.com; 2Department of Biotechnology and Bioinformatics, Korea University, Sejong 30019, Republic of Korea; jklehdgus@korea.ac.kr

**Keywords:** calcium carbonate, vaterite, chiral toroid, aspartic acid, sonochemical synthesis

## Abstract

The metastable vaterite polymorph of calcium carbonate (CaCO_3_) holds significant practical importance, particularly in regenerative medicine, drug delivery, and various personal care products. Controlling the size and morphology of vaterite particles is crucial for biomedical applications. This study explored the synergistic effect of ultrasonic (US) irradiation and acidic amino acids on CaCO_3_ synthesis, specifically the size, dispersity, and crystallographic phase of curved-edge vaterite with chiral toroids (chiral-curved vaterite). We employed 40 kHz US irradiation and introduced L- or D-aspartic acid as an additive for the formation of spheroidal chiral-curved vaterite in an aqueous solution of CaCl_2_ and Na_2_CO_3_ at 20 ± 1 °C. Chiral-curved vaterites precipitated through mechanical stirring (without US irradiation) exhibited a particle size of approximately 15 μm, whereas those formed under US irradiation were approximately 6 μm in size and retained their chiral topoid morphology. When a fluorescent dye was used for the analysis of loading efficiency, the size-reduced vaterites with chiral morphology, produced through US irradiation, exhibited a larger loading efficiency than the vaterites produced without US irradiation. These results hold significant value for the preparation of biomimetic chiral-curved CaCO_3_, specifically size-reduced vaterites, as versatile biomaterials for material filling, drug delivery, and bone regeneration.

## 1. Introduction

The fundamental phenomenon of chirality pervades the natural world, manifesting in everything from the atomic arrangement of amino acids to the macroscopically helical tooth [1,2]. The biomineralization of calcium carbonate, observed in the hardened structures of marine and terrestrial invertebrates, including helical gastropod shells and extinct ammonites, exemplifies the prevalence of chirality in biology [3,4,5,6,7,8]. In biology, the biomineralization of calcium carbonate polymorphs and the distinctive hand-feel of chiral structures are believed to be orchestrated by the action of chiral biomolecules [2]. However, knowledge of how chiral molecules guide nanoscale calcium carbonate ‘building blocks’ to form larger chiral hierarchies remains elusive.

Calcium carbonate has three polymorphs: calcite with rhombohedral symmetry, aragonite with orthorhombic symmetry, and vaterite with hexagonal symmetry [9,10,11]. Thermodynamically, vaterite represents the least stable phase; accordingly, it rarely occurs as a geological mineral in sediments [12]. Despite being metastable, vaterite is still found in nature, such as in mineral springs, often appearing as a minority component of a larger structure or as a result of a pathological process in humans and animals. Vaterite is present in fish otoliths, freshwater pearls, healed scars of some mollusk shells, gallstones, and urinary calculi [13]. In these circumstances, impurities, such as metal ions [14] or organic matter [15], may stabilize vaterite, preventing its transformation into calcite or aragonite.

Compared to calcite and aragonite, vaterite is distinguished by its high specific surface area, porosity, solubility, and spherical shape, all of which facilitate its use [16,17,18]. Simultaneously, within the field of material science, vaterite is a highly appealing CaCO_3_ polymorph with potential applications in implant design, regenerative medicine, bone implants, targeted drugs, and personal care products [16,19,20]. Mesoporous CaCO_3_ crystals with sizes in the range of 3–15 μm have attracted significant scientific attention as fully decomposable biocompatible drug delivery vectors [21,22]. Furthermore, vaterite microparticles of a certain size (0.2–4 μm) have been reported to improve the cleaning and abrasive characteristics of dentifrice formulators, teeth whitening products, and mouthwashes [16]. Vaterite also serves as a sacrificial template in the production of microcarriers and scaffolds for tissue engineering under biologically friendly conditions [23]. Moreover, promising endeavors involving in vitro and in vivo drug delivery using microcapsules templated on vaterite particles have been demonstrated, showcasing the application of vaterite in disease irradiation and vaccination [16,24,25].

The in vitro initiation of vaterite formation by soluble biomimetic molecules has spurred extensive studies on the mechanisms of its nucleation, growth, and stabilization, aiming to refine the overall understanding of the CaCO_3_ biomineralization process [19,20,26,27,28,29,30]. Biomineralization is a process that occurs within most organisms and is characterized by the regulation of crystal growth by proteins rich in acidic amino acid residues. Notably, acidic amino acids (Asp and Glu) have been implicated in biochemical homochirality [31,32]. Additionally, organic and ionic additives play crucial roles in modifying and/or controlling the crystallization of the three CaCO_3_ polymorphs. Examples of additives studied to date include amino acids [33], polyelectrolytes [34,35,36], polypeptides [37], polycarboxylic acids [38], and foreign ions [39,40].

The polymorphs of CaCO_3_ can also be modified through ultrasonic (US) irradiation. While the complete effects of US irradiation on CaCO_3_ polymorphism are not yet fully understood, literature reports indicate that US irradiation enhances the spontaneous precipitation of CaCO_3_ [41,42]. Ultrasound-assisted crystallization, or sonochemical crystallization, involves nuclei formation due to a rapid cavitation process that reduces the growth of crystals [43,44]. Sonochemical crystallization augments the number of nuclei and alters the primary nucleation growth rate, modifying the shape, size, and distribution (uniformity) of particles [45,46]. While some studies have employed US irradiation for vaterite generation, they have not yet explicitly addressed the crucial aspect of simultaneously controlling both morphology and size. Moreover, there is no documented research on the synergistic application of US irradiation with acidic amino acids to achieve precise control over the size and chiral properties of CaCO_3_ polymorphs.

In the present study, we focused on the effect of US irradiation on calcium carbonate synthesis, specifically examining its influence on the size, dispersity, and crystallographic phase of curved-edge vaterite with chiral toroids (chiral-curved vaterite) produced in the presence of acidic amino acids. Chiral-curved vaterite was synthesized using L- and D-aspartic acids at adjusted concentrations. Additionally, we sought to elucidate the mechanism through which ultrasound influences vaterite formation by varying the duration of irradiation. The effect of amino acid and US irradiation on the controlled formation of chiral-curved vaterites with specific sizes demonstrated the ability to manage the particle morphology of chiral-curved vaterites, showcasing their potential for biomaterial applications.

## 2. Materials and Methods

### 2.1. Materials

Materials and reagents, including calcium chloride dehydrate (CaCl_2_), L-aspartic acid, and D-aspartic acid, were purchased from Sigma Aldrich (St. Louis, MO, USA). Sodium carbonate anhydrous (Na_2_CO_3_) was procured from Duksan Pure Chemicals (Ansan, Republic of Korea), and commercially pure titanium (Grade 2) was sourced from Koralco Corporation (Gwangju, Republic of Korea). Additionally, 35% HCl and 95% H_2_SO_4_ were purchased from Samchun (Pyeongtaek, Republic of Korea). All other chemical reagents utilized were of analytical grade.

### 2.2. Acid-Etching Irradiation of Titanium Surface

A commercially pure Ti plate (Grade 2) was cut into 10 mm × 10 mm × 0.1 mm pieces and washed using a US cleaner (Branson Ultrasonics, Danbury, CT, USA) containing ethanol and deionized water. For preparing roughened Ti (rTi), the Ti surface was subjected to acid etching with a mixed solution of 95% H_2_SO_4_ and 35% HCl in a 1:1 ratio at 70 °C for 15 min. Subsequently, the rTi was washed thoroughly using the US cleaner containing ethanol and deionized water.

### 2.3. Preparation of Curved-Edge Vaterite with Chiral Toroid Calcium Carbonate Particles

In a 250 mL beaker, 50 mL of 3 mM Na_2_CO_3_ and 50 mL of 3 mM CaCl_2_ solution containing 20 mM L-aspartic acid or D-aspartic acid are mixed. Calcium carbonate was formed by adjusting the pH of the solution mixture to 10.5 ± 0.2 through the stepwise addition of 5 M NaOH or HCl at 20 ± 1 °C with 250 rpm stirring. The final concentrations of CaCl_2_, Na_2_CO_3_, and amino acids were 1.5, 1.5, and 10 mM, respectively. The control experiment was achieved by following the same procedures except for the use of L- or D-aspartic acid. The final concentrations of CaCl_2_ and Na_2_CO_3_ were 1.5 and 1.5 mM, respectively. A slide glass or etched Ti film was gently placed at the bottom of the beaker. Subsequently, US radiation was applied at a frequency of 40 kHz for 90 s, and mineral growth was carried out at 20 ± 1 °C for 1 to 8 h (Figure 1). After the set time reaction, the slide glass or etched Ti film with grown calcium carbonate particles was moved from the reactant beaker and rinsed with deionized water and ethanol. Then, the samples on the plates were dried in an oven at 60 °C and used for the analysis (Figure 1). Particle samples were separately removed from the slide glass or etched Ti plates and used for the subsequent experiments.

### 2.4. Characterization of Vaterite Calcium Carbonate

#### 2.4.1. Scanning Electron Microscopy (SEM) Analysis

The morphology of calcium carbonate vaterite was determined using scanning electron microscopy (SEM; JSM-5600, JEOL, Tokyo, Japan). All samples were coated with platinum using an ion coater (KIC-1A, COXEM, Daejeon, Republic of Korea).

#### 2.4.2. Attenuated Total Reflectance Fourier Transform Infrared (ATR-FTIR) Analysis

The ATR-FTIR analysis was performed at the Korea Basic Science Institute (KBSI; Daegu, Republic of Korea) using an infrared spectrophotometer (Vertex-80/Hyperion-3000) in transmittance mode over a spectral range of 600–4000 cm**^−^**^1^ and a resolution of 2 cm**^−^**^1^.

#### 2.4.3. X-ray Diffraction (XRD) Analysis

The wide-angle XRD measurements were conducted at the Korea Basic Science Institute (KBSI; Daegu, Republic of Korea) using a Panalytical Empyrean diffractometer with Cu-Kα radiation (λavg = 1.5425 Å), operating at 40 kV and 25 mA. The diffraction spectra of powder samples were collected from 10° to 80° of 2θ at a scan rate of 0.04°/s. The CaCO_3_ phase analysis was calculated using the OriginPro 10.1 software [47,48].

#### 2.4.4. Fluorescence Analysis

A fluorescent substance, specifically Rhodamine 6G, was loaded into the obtained chiral-curved vaterite to evaluate their loading efficiency. The adsorption method was used. Briefly, 10 mg of each type of dried CaCO_3_ particle was incubated in 0.1 mg/mL Rhodamine 6G aqueous solution at 20 ± 1 °C for 1.5 h with shaking (250 rpm). Then, the particles were centrifuged at 5000× *g* for 30 s, and the supernatants were removed. The fluorescence of the precipitate was analyzed through confocal laser scanning microscopy using a BX41 microscope (Olympus, Tokyo, Japan) and a STEDYCON (Abberior Instruments, Göttingen, Germany). To visualize Rhodamine 6G inside CaCO_3_ particles, a 488 nm laser was used for luminescence excitation. The integrated intensity (Z coordinate) of the vaterite staining was analyzed using ImageJ’s interactive 3D surface plotting technique for image data. Pixels with higher intensity values were located higher on the Z-axis. The specific intensity of the fluorescent-stained CaCO_3_ was calculated using the following formula: the specific intensity = total sum of the integrated intensity of fluorescent signal/total area of stained CaCO_3_.

### 2.5. Statistical Analysis

Data are presented as the mean ± standard deviation (*n* = 6). Statistical difference was analyzed using one-way analysis of variance (ANOVA). To ensure the accuracy of the data, three quantitative measurements were taken for each group, and the differences between the vaterite particles produced with and without US irradiation were determined using a standard Student’s *t*-test (*p* > 0.05, not significant; 0.01 ≤ *p* < 0.05, statistically significant; *p* < 0.01, statistically very significant).

## 3. Results and Discussion

### 3.1. Morphological Analysis

The polymorph formed during the precipitation of CaCO_3_ depends on various experimentally controllable conditions, including the degree of concentration, pH, temperature, and the presence of additives (e.g., amino acid or other metal ions). Figure 2 illustrates the morphological changes in vaterite under the influence of amino acids and US irradiation. Amino acids promoted the formation of curved-edge vaterite with chiral torotid morphology (chiral-curved vaterite), while US irradiation led to the formation of smaller chiral-curved vaterite. US irradiation mediated the formation of the vaterite phase, surpassing the formation of both the calcite and aragonite forms [49]. The addition of acidic amino acids resulted in the formation of chiral-curved vaterite. On the other hand, when a combined irradiation of amino acids and US irradiation (120 s) was applied, chiral-curved vaterite was formed in large quantities. Compared to the no-use of US irradiation, approximately 20% more chiral-curved vaterite were formed. Also, chiral-curved vaterite formed with precipitation times up to 4 h showed differences in formation with and without US irradiation. Thus, chiral-curved vaterite can be produced in large quantities by affecting nucleation, with or without sonication. This effect can be attributed to the disagglomeration of nucleation sites during the recrystallization process [50]. Chiral-curved vaterite is induced through the intermediate chelated by Asp and calcium ion, and then Asp is adsorbed on the surface of vaterite by chelation, preventing the metastable vaterite from transforming to calcite via a dissolution−recrystallization process [30].

US irradiation delivers a substantially high amount of energy through the implosion of cavitation bubbles, enhancing mass transfer and expediting chemical reactions [42,51,52]. Additionally, it produces mechanical fluctuation and thermal effects, generating microdisturbances in the supercritical phase [42,53]. The resulting turbulence decreases the boundary layer, thereby increasing the mass transfer rate. A noteworthy aspect of this process is the potential for a tenfold increase in vaterite production without significant losses in the sample quality and size, facilitating the scaling up of the synthesis procedure to an industrial level. Additionally, the reaction time has a strong influence on the size and uniformity of CaCO_3_ particles [54]. Overall, US irradiation at the early stage creates a favorable environment for the formation of chiral-curved vaterite. Furthermore, the localized increase in ion concentration in the reaction mixture, driven by the heterogeneity of its distribution, can accelerate vaterite–calcite recrystallization, leading to the formation of almost 30% calcite in the sample (due to incomplete diffusion). Conversely, homogenizing the reaction mixture using US irradiation might facilitate the formation of a vaterite phase.

Figure 3 illustrates the influence of US irradiation on the size and phase of fabricated chiral-curved vaterite. In the control sample stirred with a magnetic bar (Figure 3A), cubic-like shaped particles representing calcite are dominant. In contrast, the products obtained in the presence of acidic amino acids predominantly contained chiral-curved vaterite. L-enantiomers of Asp induced the production of chiral-curved vaterite with toroidal suprastructure exhibiting a counterclockwise spiraling morphology, whereas D-enantiomers induced the production of vaterite with structures of a clockwise morphology. Moreover, sequential switching between the amino acid enantiomers resulted in a corresponding switch in chiral toroids (Figure 3B,C) [2,55]. These results demonstrate that the chiral morphology and hierarchically organized architecture of vaterite can be controlled by adding chiral acidic amino acids (e.g., L-Asp and D-Asp). When the samples were subjected to US irradiation in the absence of aspartic acid, circular vaterite particles in the submicron size range were produced (Figure 3D). Interestingly, combined irradiation involving both US irradiation and the presence of aspartic acid resulted in a reduction in particle size of chiral-curved vaterite and retention of the chiral morphology shape (Figure 3E,F).

The precipitation of CaCO_3_ in the presence of amino acids, particularly acidic amino acids, has been extensively investigated due to its significance in diverse fields, including crystallization, biomineralization, and geology [27,56,57]. In previous works, the induction of symmetric/achiral vaterite using acidic amino acids has been demonstrated [2,30,55,58]. However, the combined use of amino acids and US irradiation has not yet been reported to date for controlling the size and properties of CaCO_3_ polymorphs. While there are some studies that have used US irradiation to form vaterite, there have been no reports to focus on controlling the morphology and size of the CaCO_3_ polymorphs at the same time.

The effect of the duration of US irradiation on the synthesis process was also investigated. Figure 4A–D depict the morphology and size of vaterite CaCO_3_ precipitated through the combined use of US irradiation and L-aspartic acid in an aqueous solution of CaCl_2_ and Na_2_CO_3_ at 25 °C. As the duration of US irradiation increased, the size of the chiral-curved vaterite particles decreased. US irradiation was performed for durations ranging from 30 s (Figure 4A) to 120 s (Figure 4D), followed by precipitation. The results revealed that increasing the duration of US irradiation reduces the size of the curved-edge vaterite particles. The size distribution of the vaterite particles decreased from 13.4 to 6.5 μm as the duration of US irradiation increased. Moreover, despite the smaller particle size, the chirality became more pronounced (Figure 4D,H). However, US irradiation for more than 120 s led to heat generation, making it challenging to observe the chiral-curved vaterite particles. Figure 4E–H illustrate the morphology and size of vaterite CaCO_3_ precipitated using D-Aspartic acid under similar conditions. The size distribution of the vaterite particles decreased from 16.9 to 9.0 μm as the duration of US irradiation increased. Notably, the use of the L-form resulted in a greater reduction in the size of the chiral-curved vaterite compared to the D-form.

Overall, sonochemical crystallization increased the number of nuclei and altered the primary nucleation growth rate, influencing particle shape, size, and distribution uniformity [59,60,61]. Furthermore, the identified US irradiation conditions effectively facilitated the synthesis of chiral-curved vaterite particles with chiral properties and minimal size.

Average particle morphology, size, percentage of vaterite production, and ratio were determined through statistical analysis of SEM images and are summarized in Table 1. US irradiation-based agitation resulted in a more homogeneous distribution of growth centers, facilitating the formation of the smallest particles (6.56 ± 0.56 μm size). While there was no difference in the chiral toroid morphology of the sonicated and nonsonicated samples, the sonicated samples exhibited a significant reduction in particle size. Furthermore, there was an increase in both the amount and proportion of chiral-curved vaterite formed. In particular, the use of L-Asp resulted in a two-fold increase in the production of chiral-curved vaterite. Thus, the optimal environmental conditions for producing chiral-curved vaterite were successfully identified.

### 3.2. XRD and FTIR Analysis

Figure 5A shows the XRD patterns of the CaCO_3_ crystals obtained in pure media and amino acid-containing media with and without US irradiation. In the 20°–40° range, the XRD patterns revealed the presence of vaterite at 24.9°, 27.1°, and 32.7° in 2θ, reflecting crystal planes (100), (101), and (102), respectively. For calcite, key diffraction peaks were observed at 23°, 29.3°, 35.9°, and 39.5° in 2θ, corresponding to crystal planes (012), (104), (110), and (113), respectively. The XRD patterns show that the phase structure of CaCO_3_ particles was distinctly influenced by the presence of acidic amino acids with or without ultrasonication, namely LA (L-Asp), DA (D-Asp), LAS (L-Asp + US irradiation), and DAS (D-Asp + US irradiation). Specifically, they indicated that the presence of amino acids promoted the formation of chiral-curved vaterite and inhibited calcite formation [33,62]. Thus, the addition of amino acids directly influences the crystal structure and promotes the formation of the chiral-curved vaterite phase.

To monitor the polymorphic transformation in CaCO_3_ crystallization in the presence of acidic amino acids, the changes in the crystal polymorphs were confirmed using ATR-FTIR. Figure 5B shows the FTIR spectra of CaCO_3_ crystals obtained in pure and amino acid-containing media with or without US irradiation. The FTIR spectra clearly demonstrated that the calcite and vaterite forms have different functional groups. The FTIR spectra revealed two characteristic peaks at 872 and 744 cm^−1^, representing the out-of-plane and in-plane bending modes of CO_3_^2−^, respectively. The FTIR characteristics indicated that pure vaterite was formed in all samples that underwent US irradiation. The bands representing doubly degenerate in-plane O-C-O deformation bending, CO_3_ out-of-plane bending, and symmetric C-O stretching in vaterite are typically located at 744 cm**^−^**^1^, 872 cm**^−^**^1^, and 1076–1089 cm**^−^**^1^, respectively. On the other hand, the bands representing doubly degenerate in-plane O-C-O deformation bending and CO_3_ out-of-plane bending in calcite are typically located at 714 and 848 cm**^−^**^1^, respectively [60,63,64].

The phase percentages of CaCO_3_ were determined utilizing OriginPro software with XRD data. XRD calcite peak analysis showed that with L-Asp, calcite accounted for 30.3% of the total CaCO_3_ phase, while after US irradiation, the calcite percentage decreased to 19.6%. Similarly, when D-Asp was used, calcite accounted for 32.1%, but after US irradiation, the calcite percentage decreased to 27.5%. Considering the relative proportion of vaterite, L-Asp and D-Asp of vaterite percentage was 69.5% and 67.5% of the total CaCO_3_ phase, while after US irradiation, the vaterite percentage was 80.1% and 72.3%, respectively. Taking into consideration all of the above findings, the combined use of acidic amino acids and US irradiation could affect the CaCO_3_ crystal phases and increase the formation of the chiral-curved vaterites.

### 3.3. Loading Efficiency of Curved-Edge Vaterite with Chiral Toroid Morpholoy

Loading efficiency is an important property for the applications of biomimetic materials. Rhodamine 6G was used to determine the loading properties of the obtained porous CaCO_3_ particles. Confocal fluorescent microscopy images illustrated the uniform distribution of Rhodamine 6G within the porous structure of the studied particles. Calcite exhibited minimal Rhodamine 6G retention (Figure 6A), whereas chiral-curved vaterite produced without US irradiation exhibited effective retention of Rhodamine 6G (Figure 6B,C). Furthermore, chiral-curved vaterite produced using US irradiation exhibited considerably higher retention of Rhodamine 6G (Figure 6D,E). Of these, chiral-curved vaterite prepared using L-Asp and US irradiation displayed superior Rhodamine 6G retention efficiency (Figure 6D).

The loading efficiency of vaterite particles is intricately linked to their particle size and morphology. Thus, in the case of calcium carbonate, there is a strong dependence on the ‘quality’ of the sample, which corresponds to the percentage of porous vaterite and smooth calcite phases in a sample. A reduced percentage of calcite and a smaller particle size contribute to enhanced loading efficiencies, thereby improving the usability of the biomimetic material. Similarly, a higher percentage of vaterite and a smaller particle size contribute to improved loading efficiencies, thereby enhancing the applicability of the biomimetic material. When dealing with low-molecular-weight substances, such as Rhodamine 6G, the loading properties remain identical across all samples because the size and external surface of the particle area of such small substances have negligible influence on their adsorption. Therefore, the particles adsorb such small substances uniformly as a bulk material [60,65].

The absorption of Rhodamine 6G staining was analyzed using a 3D interactive surface plot [65,66,67,68]. Figure 7A illustrates that, in comparison to calcite (control), vaterite exhibited higher Rhodamine 6G absorption. The maximum intensity projection (Z coordinate) revealed that the vaterite produced through US irradiation (Figure 7D,E) exhibited higher fluorescence compared to the vaterite produced without US irradiation (Figure 6B,C). Additionally, the specific intensity is determined by the sum of the integrated intensity of the fluorescent signal per total area of each CaCO_3_ type (Figure 7F). The curved vaterite produced through US irradiation exhibited higher specific intensity compared to the curved vaterite produced without US irradiation. Therefore, consistent with the previous finding, chiral-curved vaterite produced through US irradiation and acidic amino acid provided more favorable morphology for small compounds. These results indicate that chiral-curved vaterites are useful biomimetic materials, as evidenced by their reduced size and increased density after US irradiation.

## 4. Conclusions

This study reported the effect of US irradiation on the synthesis of curved-edge vaterite with chiral toroid morphology (chiral-curved vaterite) in the presence of an acidic amino acid. Furthermore, it compared the effects of US agitation and magnetic stirring of the reaction solution, focusing on the morphology of the resulting vaterite particles, their size dispersity, crystallographic phase, and reaction yield. The results demonstrated that ultrasound-assisted precipitation improved calcium and carbonate ion distribution, leading to the rapid formation of chiral-curved vaterite. Furthermore, analyses of the SEM images and XRD spectra of the samples revealed that the use of US irradiation and amino acids increased the proportion of particles in the vaterite phase during the early stage. Overall, this study demonstrated that ultrasound-assisted precipitation can serve as a green method for controlling the morphology, polymorph type, and crystal size of chiral-curved vaterite minerals. To the best of our knowledge, this is the first report on the combined use of US irradiation and acidic amino acids for the controlled synthesis of chiral-curved vaterite. These results, obtained by employing US waves and amino acids in inorganic phase synthesis, hold promise for the development of new strategies for creating novel biomimetic materials with desirable shapes and properties.

## Figures and Tables

**Figure 1 biomimetics-09-00174-f001:**
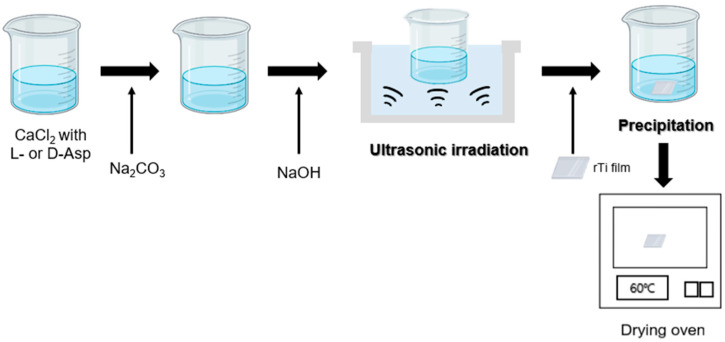
The schematic diagram and process of curved-edge vaterite preparation.

**Figure 2 biomimetics-09-00174-f002:**
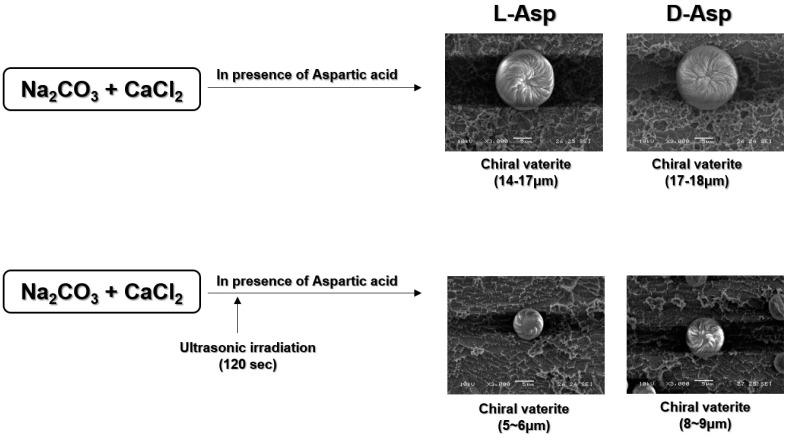
Scheme of fabrication of curved-edge vaterite with chiral toroid with or without ultrasonication in the presence of acidic amino acids. The size-decreased formation of spheroidal vaterite particles in an aqueous solution of CaCl_2_ and Na_2_CO_3_ at 20 ± 1 °C temperature, with 40 kHz sonication and the presence of L- or D-Aspartic acid as an additive.

**Figure 3 biomimetics-09-00174-f003:**
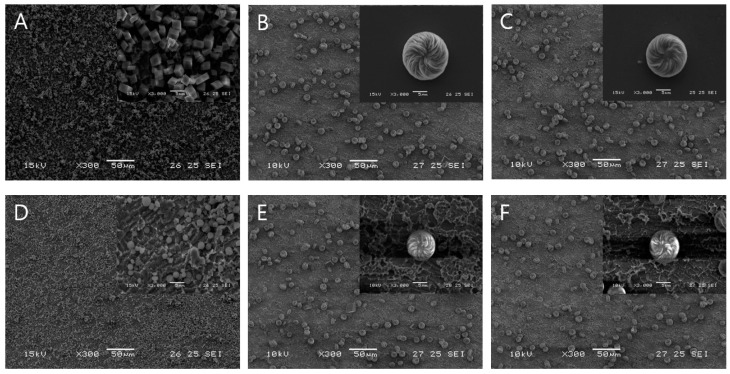
Morphology and size of CaCO_3_ precipitated with and without amino acids and US irradiation in an aqueous solution of CaCl_2_ and Na_2_CO_3_ at 20 ± 1 °C. Scanning electron microscope (SEM) images of CaCO_3_ produced without US irradiation; (**A**) calcite produced in the absence of amino acids, (**B**) chiral-curved vaterite produced in the presence of L-Asp, and (**C**) chiral-curved vaterite produced in the presence of D-Asp. SEM images of CaCO_3_ obtained after US irradiation: (**D**) the mixture of calcite and vaterite produced in the absence of amino acids, (**E**) chiral-curved vaterite produced in the presence of L-Asp, and (**F**) chiral-curved vaterite produced in the presence of D-Asp. Small-scale bars represent 5 μm and large-scale bars 50 μm.

**Figure 4 biomimetics-09-00174-f004:**
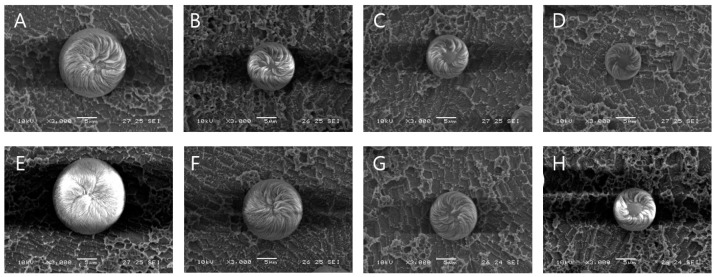
The effect of US irradiation on the morphology and size of vaterite CaCO_3_ precipitated in the presence of acidic amino acids (as an additive) in an aqueous solution of CaCl_2_ and Na_2_CO_3_ at 20 ± 1 °C. SEM images of vaterite produced in the presence of L-Asp and 30 s (**A**), 60 s (**B**), 90 s (**C**), and 120 s (**D**) of US irradiation. SEM images of vaterite produced in the presence of D-Asp and 30 s (**E**), 60 s (**F**), 90 s (**G**), and 120 s (**H**) of US irradiation. All scale bars represent 5 μm.

**Figure 5 biomimetics-09-00174-f005:**
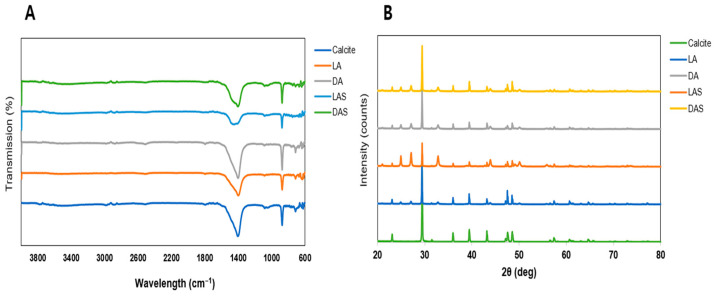
Fourier transform infrared (FT-IR) spectra (**A**) and X-ray diffraction (XRD) (**B**) of the CaCO_3_ particles synthesized under the control (without US irradiation and additives) and irradiation (US irradiation and addition of Asp as an additive) conditions. Chiral-curved vaterite produced in the presence of L-Asp called LA, or in the presence of D-Asp, called DA. Ultrasound was treated for chiral-curved vaterite produced in the presence of L-Asp and D-Asp, called LAS and DAS, respectively.

**Figure 6 biomimetics-09-00174-f006:**
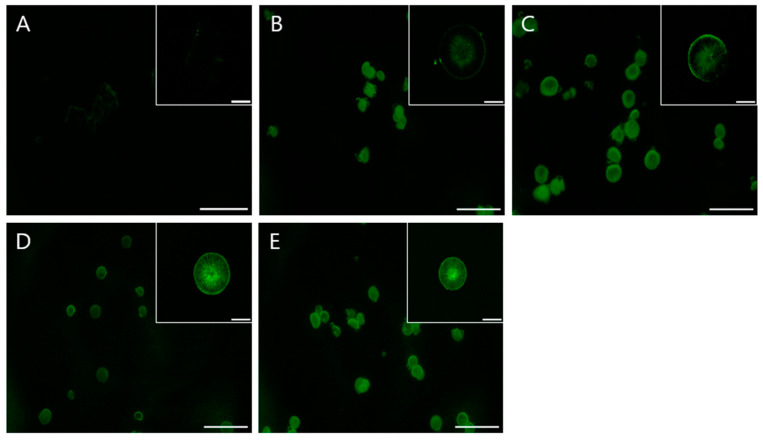
Confocal microscopy images of calcite (**A**) and chiral-curved vaterite (produced using L-Asp and D-Asp) loaded with Rhodamine 6G. (**B**,**C**) Chiral-curved vaterite produced in the presence of L-Asp and D-Asp, respectively. (**D**,**E**) Chiral-curved vaterite produced through irradiation with US radiation in addition to using L-Asp and D-Asp, respectively. Small-scale bar = 5 µm. Large-scale bar = 100 µm.

**Figure 7 biomimetics-09-00174-f007:**
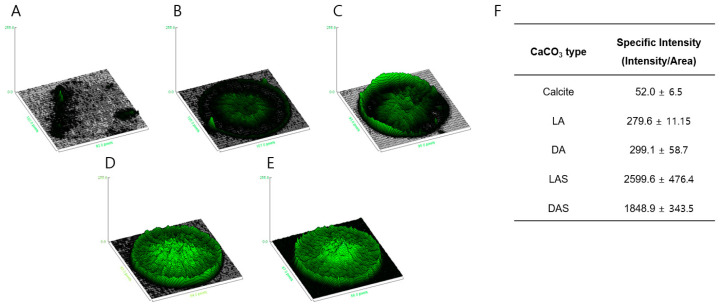
Interactive 3D surface plots displaying pixel intensity (Z) distributions of calcite and chiral-curved vaterite (produced in the presence of L-Asp or D-Asp) in grayscale images; (**A**) calcite (control), (**B**) chiral-curved vaterite produced in the presence of L-Asp, called LA, and (**C**) chiral-curved vaterite produced in the presence of D-Asp, called DA. (**D,E**) Ultrasound-treated chiral-curved vaterite produced in the presence of L-Asp and D-Asp, called LAS and DAS, respectively. (**F**) The specific intensity is determined by considering the sum of integrated intensity of fluorescent signal per total area of each CaCO_3_ type.

**Table 1 biomimetics-09-00174-t001:** Particle properties of chiral-curved vaterite produced under different irradiation conditions.

Additive	Ultrasonic Irradiation	Morphology	Chiral-Curved Vaterite Size (μm)
L-Asp	-	Platelets, Helicoid	13.40 ± 0.63
D-Asp	-	Platelets, Helicoid	16.98 ± 0.87
L-Asp	90 s	Platelets, Helicoid	6.56 ± 0.56
D-Asp	90 s	Platelets, Helicoid	9.0 ± 0.61

## Data Availability

Dataset available on request from the authors.

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
