# Peer review of "Size Control of Biomimetic Curved-Edge Vaterite with Chiral Toroid Morphology via Sonochemical Synthesis"

_biomimetics, 2024, doi:10.3390/biomimetics9030174_

Round 1

Reviewer 1 Report

Comments and Suggestions for Authors

2.3. Preparation of chiral vaterite calcium carbonate particles. It is not clear if the microparticles of CaCO3 are obtained just on the glass or Ti surfaces or also in bulk. I do not understand how the 10 mg of microparticles for fluorescence studies were obtained. Please clarify the methods part. Also, I may assume that the formation of CaCO3 microparticles took place immediately after mixing the CaCl2 (with or without amino acid) and NaCO3. Why the glass or Ti surfaces were introduced after components mixing, and just then US was applied? The stirring information should be also added. After US how the “mineral growth” was done?

The scheme included in Figure 1 is not in agreement with its legend. No any information really connected to the calcite-to-vaterite phase transition. Also, the same SEM images are also found in Figure 3 E, F. Thus, I consider that this figure can be removed.

The polymorphs composition is expected to be obtained by FTIR and/or XRD. Thus, Figure 2 and the corresponding discussion should be after the description of that spectra/diffractograms. Or, please explain how the experimental mass yield of each polymorph was determined. The polymorphs % was determined on the CaCO3 formed in bulk or on the glass or Ti surface? Please check the Fig 2 legend: “amino acid and ultrasonic irradiation (A), ...... mixing with amino acid and ultrasonic irradiation (C)” - A and C seems identical

Usually, during aging vaterite is transformed into the most stable form, calcite. How authors can explain the decrease of calcite content in the 8 h investigation, with or without US (Fig 2)?

Figure 3. Even if the already published studies shows that achiral vaterite is formed with different aminoacids (line 219), herein chiral vaterite was formed, with almost the same vaterite % without US (Figure 2 and 3). How this can be explained? We may assume all the SEM images in Fig 3 are after 8h aging, with or without US?

The temperature was change to 25oC in Figures 3 and 4 (in 2.3 part is 20 oC). Why?

“the L-form resulted in a greater reduction in the size of the chiral vaterite compared to the D-form.” Why?

How can be determined the percentage of vaterite production and ratio by analysis of SEM images? How this method can discriminate among the CaCO3 polymorphs? How can SEM give the mg/100 mL production?!

Figure 5 – A is FTIR and B is XRD. Also, WRD is enough in the range 20-60 2theta, and FTIR in the range 1800-400 cm-1? Why the same “called” samples in Figs 5 were not abbreviated the same in the whole manuscript? How calcite was obtained and why included in the Figure 5? Only sample LAS clearly show a decrease in calcite content but still a very large calcite content – not in agreement with Fig 2.

“Moreover, the XRD patterns demonstrated that only vaterite crystals are produced when amino acids are included in the reaction mixture, irrespective of the presence or absence of US treatment.” It is not true. Please see the intense peak at 29.3 2 theta. DAS diffractogram is almost identical with calcite, just some very small diffraction peaks assigned to vaterite being visible.

What means “The FTIR spectra clearly demonstrated that the calcite and vaterite forms have different functional groups.” Both polymorphs have the same chemical structure! Also, since the spectra of calcite is almost identical the other samples, it is not correct to say that “The FTIR characteristics indicated that pure vaterite was formed in all samples that underwent US treatment”. Maybe if the authors will see the spectra enlarging the 1800 – 600 cm -1 range will observe better. I suggest to follow the peaks at 745 cm -1 (vaterite) and (713 cm-1 (calcite) and at 1087 cm-1 (absent in calcite). No any peaks assigned to the aminoacids are seen?

The presence of aminoacids in the formed microparticles have no any influence on the rhodamine sorption? The images in Fig 6 show that rodamine is not uniform distributed in the microparticles. The loading capacity is usually expressed as an amount of loaded component versus the sorbent material. I consider inappropriate the usage of this term in relation with the presented results

Even if the term “chiral” is used in the whole manuscript no any chirality proof is shown. It is just an assumption based on the arrangements of the nanocrystals in the formed composite structures (containing aminoacids and CaCO3).

Author Response

# 2.3. Preparation of curved vaterite calcium carbonate particles. It is not clear if the microparticles of CaCO3 are obtained just on the glass or Ti surfaces or also in bulk. I do not understand how the 10 mg of microparticles for fluorescence studies were obtained. Please clarify the methods part.

Answer:

Thanks for your comments. We generated vaterite particles on the surfaces and removed them from the surface for the subsequent experiments. In the revision, we mentioned how to obtain vaterite on section 2.3.

# Also, I may assume that the formation of CaCO3 microparticles took place immediately after mixing the CaCl2 (with or without amino acid) and NaCO3. Why the glass or Ti surfaces were introduced after components mixing, and just then US was applied? The stirring information should be also added. After US how the “mineral growth” was done?

Answer:

The plates were put in after the pH adjustment. Then, US treatment was employed according to the report of Y.I. Svenskaya et al (Advanced Powder Technology 27 (2016) 618-624), For consistent nucleation, the plate surface was put before the US treatment. After US treatment, crystal nuclei on the surface grows into vaterite. In the revision, we added information of stirring method in section 2.3.

# The scheme included in Figure 1 is not in agreement with its legend. No any information really connected to the calcite-to-vaterite phase transition. Also, the same SEM images are also found in Figure 3 E, F. Thus, I consider that this figure can be removed.

Answer:

We made mistakes. We edited the legends. Figure 1 showed the scheme for the use of acidic amino acid and US treatment for the size-decreased formation of vaterite particles with the chiral toroid maintained. We modified Figure 1 accordingly.

# The polymorphs composition is expected to be obtained by FTIR and/or XRD. Thus, Figure 2 and the corresponding discussion should be after the description of that spectra/diffractograms. Or, please explain how the experimental mass yield of each polymorph was determined. The polymorphs % was determined on the CaCO3 formed in bulk or on the glass or Ti surface? Please check the Fig 2 legend: “amino acid and ultrasonic irradiation (A), ...... mixing with amino acid and ultrasonic irradiation (C)” - A and C seems identical.

Answer:

The portion of curved vaterite was calculated by counting the vaterite forms from the low magnification SEM image of the generated CaCO3 on the plate surfaces. We edited Figure 2 legends more clearly.

# Usually, during aging vaterite is transformed into the most stable form, calcite. How authors can explain the decrease of calcite content in the 8 h investigation, with or without US (Fig 2)?

Answer:

Thanks for your comments. We don’t know accurate mechanisms to explain the results. However, we used the condition to allow the amino acid (L- or D-Asp) to get involved in the formation of curved vaterites with chiral toroid. Amino acids are known to play a role not only in the nucleation of crystals (at the early stage), but also the growth of crystals (leading to the final structure and morphology). 

Anyway, such an involvement of acidic amino acid in the growth might be accumulated as time have elapsed, so that the portion of curved vaterite with chiral toroid is getting larger, so that portion of calcite looks decreased.

# Figure 3. Even if the already published studies shows that acurved vaterite is formed with different aminoacids (line 219), herein curved vaterite was formed, with almost the same vaterite % without US (Figure 2 and 3). How this can be explained? We may assume all the SEM images in Fig 3 are after 8h aging, with or without US?

Answer:

Thanks for your comments. As the reviewer pointed out that the portion of generated vaterite is similar with or without US in the presence of acidic amino acids. The iiradiation of US is done shortly at the early stage. Such a short treatment of US can lead to reduction of particle size (below 10 micro) without any influence on the chiral toroid morphology of vaterite. In fact, the main focus of this work is to control, reduce the size of curved vaterite particles with chiral toroid.

# The temperature was change to 25oC in Figures 3 and 4 (in 2.3 part is 20 oC). Why?

Answer:

We made mistakes. We edited the values.

# “the L-form resulted in a greater reduction in the size of the curved vaterite compared to the D-form.” Why?

Answer:

Thank you for your comment.

We don’t know accurate mechanisms to explain the results. The difference may result from the effects of the amino-acid chirality. The size of curved vaterite resulted from L-Asp is observed smaller than that of D-Asp. The difference is one of interesting topics, which is our next research work.

# How can be determined the percentage of vaterite production and ratio by analysis of SEM images? How this method can discriminate among the CaCO3 polymorphs? How can SEM give the mg/100 mL production?!

Answer:

As mentioned above, the portion of curved vaterite was calculated by counting the vaterite forms from the low magnification SEM image of 100 the generated CaCO3 on the plate surfaces.

As shown in Figure 2, most of the generated CaCO3 in 8 hr is vaterite, so that we collected them upto 10 mg by removing vaterite particles from the plate surface after 8 ha incubation. 

# Figure 5 – A is FTIR and B is XRD. Also, WRD is enough in the range 20-60 2theta, and FTIR in the range 1800-400 cm-1? Why the same “called” samples in Figs 5 were not abbreviated the same in the whole manuscript? How calcite was obtained and why included in the Figure 5? Only sample LAS clearly show a decrease in calcite content but still a very large calcite content – not in agreement with Fig 2.

Answer:

We made mistakes. In revision, we did XRD and FT-IR analysis again to prepare the Figure 5A and B for clarity. Also, we editeded range of XRD and FT-IR and Abbreviated name.

# “Moreover, the XRD patterns demonstrated that only vaterite crystals are produced when amino acids are included in the reaction mixture, irrespective of the presence or absence of US treatment.” It is not true. Please see the intense peak at 29.3 2 theta. DAS diffractogram is almost identical with calcite, just some very small diffraction peaks assigned to vaterite being visible.

Answer:

We made mistakes. We deleted the sentence for clarity.

# What means “The FTIR spectra clearly demonstrated that the calcite and vaterite forms have different functional groups.” Both polymorphs have the same chemical structure! Also, since the spectra of calcite is almost identical the other samples, it is not correct to say that “The FTIR characteristics indicated that pure vaterite was formed in all samples that underwent US treatment”. Maybe if the authors will see the spectra enlarging the 1800 – 600 cm -1 range will observe better. I suggest to follow the peaks at 745 cm -1 (vaterite) and (713 cm-1 (calcite) and at 1087 cm-1 (absent in calcite). No any peaks assigned to the aminoacids are seen?

Answer:

Thank you for your comment. We made mistakes. We edited for clarity.

# The presence of aminoacids in the formed microparticles have no any influence on the rhodamine sorption? The images in Fig 6 show that rodamine is not uniform distributed in the microparticles. The loading capacity is usually expressed as an amount of loaded component versus the sorbent material. I consider inappropriate the usage of this term in relation with the presented results

Answer:

Thank you for your comment. We revised the term from "loading capacity" to "loading efficiency" for clarity.

# Even if the term “chiral” is used in the whole manuscript no any chirality proof is shown. It is just an assumption based on the arrangements of the nanocrystals in the formed composite structures (containing aminoacids and CaCO3).

Answer:

Thank you for your comment. We agree with your opinion. In revision, we considered the previous reports and specified the terminologies in the whole manuscript. For example, “Curved-edge vaterite with chiral toroid” instead of chiral-curved vaterite.

Reviewer 2 Report

Comments and Suggestions for Authors

The presented research is quite interesting and shows the precipitation of chiral vaterite under US treatment where L-enantiomers of Asp induced the production of a chiral vaterite structure exhibiting a ‘right-handed’ (counterclockwise) spiraling morphology, whereas D-enantiomers induced the production of structures with a ‘left-handed’ (clockwise). However, additional experiments and comprehensive revisions of the manuscript are necessary to enhance the overall quality.

 In the Abstract, you state the following:" These results hold significant value for the preparation of biomimetic calcium carbonate, specifically vaterites, as versatile biomaterials for material filling, drug delivery, and bone regeneration." The same/similar is stated throughout the entire introduction. The experiments were done on titanium plates, and this is more relevant for implants. If the goal is the production of vaterite for implants, more literature should be given on bone regeneration and implants. The authors are writing about drug delivery and material filing, but the experimental setup is not suitable for such applications. For drug delivery, higher amounts of precipitate are needed. It should be checked if the same results can be obtained with higher concentrations of  Na2CO3, CaCl2, and amino acid solutions. Precipitation of calcium carbonate will not be the same at lower and higher supersaturations, so it will be necessary to modify the reaction conditions. If drug delivery is the goal, then additional experiments are needed to demonstrate under what conditions the same vaterite morphology will be obtained but with a much higher reaction yield. Align the experimental setup, research aim, and introduction to convey a cohesive story. Perhaps it is best to omit the filling and drug delivery material or not emphasize it so much because the experimental setup is not suitable for such research.

Introduction:

1.       Add reference in lines 43-45.

2.       What do you mean by "as well as low specific gravity" in line 47?

3.       In vitro” and “in vivo” should be in italics throughout the manuscript.

Experimental:

1.       Please write the sentence in lines 107-109 more clearly. Did you adjust the pH of each solution before mixing, or was the pH adjusted after mixing Na2CO3 and CaCl2 solutions?

2.       If the pH of the solution was adjusted after mixing, was the solution stable without CaCO3 precipitation?

3.       Provide the initial vaterite supersaturation in the mixed Na2CO3 and CaCl2 solution.

4.       Specify the number of repetitions for the precipitation experiments.

5.       Regarding the choice of slide glass and titanium plates, address their relevance to the research goal. Consider experiments with higher Na2CO3, CaCl2, and amino acid concentrations for drug loading to assess varying conditions for consistent vaterite morphology.

6.       The sentence in lines 108-111 is contradictory. "In control experiments, where no amino acid was added and the same amount of L-aspartic acid and D-aspartic acid was present, a mixture of calcium carbonate crystals, comprising achiral rhombohedral calcite and hexagonal vaterite crystals, was obtained." Were the amino acids used in the control experiment or not? Move morphological and polymorph composition details to the Results section.

7.       Lines 152-153.: "The surface area rate of vaterite was analyzed using ImageJ software. The coverage rate was calculated using the following formula: coverage rate = area of attached fouling organism/total view field × 100%." Explain the analysis of vaterite surface area in lines 152-153. Clarify which fouling organisms were used for the analysis.

8.       Did you follow the reaction potentiometrically (pH vs time)? Is there an influence of amino acids on the inhibition of CaCO3 nucleation?

Results

1.       Change "e,g.," to e.g., in line 160

2.       Describe the calculation procedure for experimental mass yield based on theoretical values in the experimental part. Specify if XRD diffractograms (Rietveld refinement) were used for calculations.

3.       Include the monitoring of polymorph composition change over time in the experimental part, detailing the procedure and sample collection times.

4.       Explain the phenomenon of calcite to vaterite transformation with prolonged precipitation time, as shown in Figures 2B and 2C.

5.       Address the issue of text being cut off in Figure 2's horizontal axis.

6.       Emphasize the novelty of the research in lines 219-220 regarding the combined use of amino acids and US treatment, and add this information to the introduction.

7.       In lines 317-323, the porosity and surface area of the prepared crystals are mentioned. Please measure the samples' porosity and surface area (B.E.T. method) and provide the data.

8.       Specify the loading capacity of crystals and provide the concentration of adsorbed Rhodamine 6G relative to the crystal's surface area.

9. Consider doing in vivo drug release experiments, e.g. antibacterial and biocompatibility with human cells 

Comments on the Quality of English Language

Moderate editing of English language required.

Author Response

#2

The presented research is quite interesting and shows the precipitation of curved vaterite under US treatment where L-enantiomers of Asp induced the production of a curved vaterite structure exhibiting a ‘right-handed’ (counterclockwise) spiraling morphology, whereas D-enantiomers induced the production of structures with a ‘left-handed’ (clockwise). However, additional experiments and comprehensive revisions of the manuscript are necessary to enhance the overall quality.

In the Abstract, you state the following:" These results hold significant value for the preparation of biomimetic calcium carbonate, specifically vaterites, as versatile biomaterials for material filling, drug delivery, and bone regeneration." The same/similar is stated throughout the entire introduction. The experiments were done on titanium plates, and this is more relevant for implants. If the goal is the production of vaterite for implants, more literature should be given on bone regeneration and implants. The authors are writing about drug delivery and material filing, but the experimental setup is not suitable for such applications. For drug delivery, higher amounts of precipitate are needed. It should be checked if the same results can be obtained with higher concentrations of Na2CO3, CaCl2, and amino acid solutions. Precipitation of calcium carbonate will not be the same at lower and higher supersaturations, so it will be necessary to modify the reaction conditions. If drug delivery is the goal, then additional experiments are needed to demonstrate under what conditions the same vaterite morphology will be obtained but with a much higher reaction yield. Align the experimental setup, research aim, and introduction to convey a cohesive story. Perhaps it is best to omit the filling and drug delivery material or not emphasize it so much because the experimental setup is not suitable for such research.

Answer:

Thanks for your comments.

Our current manuscript is focusing on the size control of curved vaterite with chiral toroid, that is to find an effective way to reduce the size of curved vaterite. Our next works may be about the applications of the size-decreased curved-edge vaterites.

Introduction:

  1. Add reference in lines 43-45.

Answer:

Thank you for your comment. We added references.

  1. What do you mean by "as well as low specific gravity" in line 47?

Answer:

Thank you for your comment. We edited for clarity.

  1. “In vitro” and “in vivo” should be in italics throughout the manuscript.

Answer:

We made mistakes. We edited them.

Experimental:

  1. Please write the sentence in lines 107-109 more clearly. Did you adjust the pH of each solution before mixing, or was the pH adjusted after mixing Na2CO3 and CaCl2 solutions?

Answer:

Thank you for your comment. We edited sentence in line 107 – 111 more clearly.

  1. If the pH of the solution was adjusted after mixing, was the solution stable without CaCO3 precipitation?

Answer:

Thank you for your comment. Due to the use of acidic amino acids, the pH of the solution is low; Nucleation and growth begins from pH 10.5 and above, so it is stable without precipitate when pH is adjusted even after mixing.

  1. Provide the initial vaterite supersaturation in the mixed Na2CO3 and CaCl2 solution.

Answer:

Thank you for your comment. We did not focus on the supersaturation conditions. We used the condition to allow the acidic amino acid (L- or D-Asp) to affect effectively the formation of each curved vaterite with different chiral toroid. Moreover, we also found out that US irradiation could be an another effector to reduce the size of curved vaterites under the same conditions.

  1. Specify the number of repetitions for the precipitation experiments.

Answer:

Thank you for your comment. We repeated all the experiments at least 3 times considering the statistical analysis. We added the number of repetitions in Figure Legends.

  1. Regarding the choice of slide glass and titanium plates, address their relevance to the research goal. Consider experiments with higher Na2CO3, CaCl2, and amino acid concentrations for drug loading to assess varying conditions for consistent vaterite morphology.

Answer:

Thank you for your comment.

The main focus of the current work is to reduce the size of curved vaterites with chiral toroid. Of course, the titanium plate was found as more favorable surface than slide glass for reducing the particle size. However, the choice of plate kinds will not be important at present, but the current paper aims to propose US irradiation as an effective treatment employed together the use of acidic amino acids for reducing the size of vaterite with the unique chiral toroid maintained. The size-decreased chiral vatertite will be useful and the detailed applications will be our next works. We have been accumulating the data for the use of curved vaterite for implant or drug delivery as the reviewer expected.

  1. The sentence in lines 108-111 is contradictory. "In control experiments, where no amino acid was added and the same amount of L-aspartic acid and D-aspartic acid was present, a mixture of calcium carbonate crystals, comprising achiral rhombohedral calcite and hexagonal vaterite crystals, was obtained." Were the amino acids used in the control experiment or not? Move morphological and polymorph composition details to the Results section.

Answer:

Thank you for your comment. We edited them for clarity.

  1. Lines 152-153.: "The surface area rate of vaterite was analyzed using ImageJ software. The coverage rate was calculated using the following formula: coverage rate = area of attached fouling organism/total view field × 100%." Explain the analysis of vaterite surface area in lines 152-153. Clarify which fouling organisms were used for the analysis.

Answer:

Thank you for your comment. We made mistakes. We edited clearly and moved them to the end of 2.4.

  1. Did you follow the reaction potentiometrically (pH vs time)? Is there an influence of amino acids on the inhibition of CaCO3 nucleation?

Answer:

Thank you for your comment.

We did not follow the reaction potentiometrically (pH vs time). Probably, amino acids might inhibit CaCO3 nucleation, but is critical for inducing the formation of curved vaterites with chiral toroid. In fact, pH is affecting the protonation/deprotonation of aspartic acid used in this experiment. The deprotonation condition (above 9.8, so that the experiment was conducted over pH 10.5) is required for generating curved vaterites. (If not, the formation of calcite is induced largely.) The influence of reaction time is another factor, and the relationships are presented in Figure 2.

Results

  1. Change "e,g.," to e.g., in line 160.

Answer:

We made a mistake. We corrected it.

  1. Describe the calculation procedure for experimental mass yield based on theoretical values in the experimental part. Specify if XRD diffractograms (Rietveld refinement) were used for calculations.

Answer:

Thanks for your comment. The portion of curved vaterite was calculated by counting the vaterite forms or other polymorphs from the low magnification SEM image of over 100 of the CaCO3 generated on plate surface.  

  1. Include the monitoring of polymorph composition change over time in the experimental part, detailing the procedure and sample collection times.

Answer.

Thanks for your comment. We included the information in line 120. We set a range of reaction time from 1 to 8 hours and took the samples at each time for the subsequent analysis (e.g. SEM).

  1. Explain the phenomenon of calcite to vaterite transformation with prolonged precipitation time, as shown in Figures 2B and 2C.

Answer:

Thank you for your comment. We added sentence in line 171.

  1. Address the issue of text being cut off in Figure 2's horizontal axis.

Answer:

We made a mistake. We corrected it.

  1. Emphasize the novelty of the research in lines 219-220 regarding the combined use of amino acids and US treatment, and add this information to the introduction.

Answer:

Thank you for your comment. We added sentence in line 222 and 77~81.

  1. In lines 317-323, the porosity and surface area of the prepared crystals are mentioned. Please measure the samples' porosity and surface area (B.E.T. method) and provide the data.

Answer:

Thank you for your comment. In fact, the goal of our current work is to find a way to reduce the size of curved vaterite. In case of rhodamine 6G adsorption, we’d like to estimate the surface characters roughly and confirm that there are no changes in the basic properties of curved vaterite, even when the size is decreased. The subsequent and wide characterization results of curved vaterites will be useful for the next our works –application of size-decreased curved vaterite particles.

  1. Specify the loading capacity of crystals and provide the concentration of adsorbed Rhodamine 6G relative to the crystal's surface area.

Answer:

Thanks for your comment. The goal of our study is to decrease the size of curved vaterite with chiral toroid, and we used rhodamine 6G to confirm whether there are no changes in the basic properties of vaterite, even when the particle size is reduced.

In Figure 7, the specific intensity (sum of integrated fluorescence intensity / totaa area of particle) was presented to estimate the loading capacity per area of chiral varterite particles. The results will be usefully employed for the design of subsequent researches, for instance, the applications as drug delivery System.  

  1. Consider doing in vivo drug release experiments, e.g. antibacterial and biocompatibility with human cells

Answer:

Thank you for your comment. As the title suggests, the focus of current work is to find an effective way to reduce the size of the curved vaterite. This paper aims to control the size of vaterite with the chiral toroid maintained. We are now accumulating the data and preparing a follow-up work dealing with the applications of curved vaterites, for example, a drug delivery system in vivo and in vitro.

Round 2

Reviewer 1 Report

Comments and Suggestions for Authors

Even if the authors made some improvements in the manuscript revised form, I still consider that there are some concerns that should be addressed.

The authors benefit of two equipment (FTIR and XRD) which are widely used in the scientific papers to determine and quantify the polymorphs content but apply an empiric method to determine how much calcite/vaterite/aragonite (!) it was obtain in the studied samples. I consider that this is mandatory to be changed. The literature is abundant on XRD method used for this purpose. Also, if easier for authors, FTIR was used in several papers (https://doi.org/10.1016/S0039-9140(02)00638-0, https://doi.org/10.1021/cg501235r). Nevertheless, I highly recommend to remove Fig 2 and to recalculate all the precents. Actually, XRD diffractograms and FTIR spectra totally disagree Fig 2 (even if the authors decided to remove the calcite diffractogram and spectra from it). Also, the values of vaterite % and mg/100 mL in Table 1 should be removed. Collecting 10 mg powder on the plate surface it is not an accurate method to quantify a polymorph in a mixture (even if it is the main polymorph obtained). The fact that some sentences were removed “for clarity” didn’t actually improved the FTIR and XRD discussion.

The authors present some results but without understanding or trying to find explanations for experimental findings. I recommend them to read more the literature and to try to find explanation on the inhibition of calcite growth during the 8h crystallization process, why the size of particles didn’t increase in the 8h aging, the possible explanation of L or D form influence.

Author Response

#1.

# Even if the authors made some improvements in the manuscript revised form, I still consider that there are some concerns that should be addressed. The authors benefit of two equipment (FTIR and XRD) which are widely used in the scientific papers to determine and quantify the polymorphs content but apply an empiric method to determine how much calcite/vaterite/aragonite (!) it was obtain in the studied samples. I consider that this is mandatory to be changed. The literature is abundant on XRD method used for this purpose. Also, if easier for authors, FTIR was used in several papers (https://doi.org/10.1016/S0039-9140(02)00638-0, https://doi.org/10.1021/cg501235r).

# Nevertheless, I highly recommend to remove Fig 2 and to recalculate all the percents. Actually, XRD diffractograms and FTIR spectra totally disagree Fig 2 (even if the authors decided to remove the calcite diffractogram and spectra from it).

Answer:

Thanks for your comments.

We removed the Figure 2 and recalculated the percentage from XRD. We added the recalculated the results from line 316.

# Also, the values of vaterite % and mg/100 mL in Table 1 should be removed.

Collecting 10 mg powder on the plate surface it is not an accurate method to quantify a polymorph in a mixture (even if it is the main polymorph obtained).

Answer:

Thanks for your comments. We removed the values of vaterite % and mg/100 mL in Table 1.

# The fact that some sentences were removed “for clarity” didn’t actually improved the FTIR and XRD discussion. The authors present some results but without understanding or trying to find explanations for experimental findings. I recommend them to read more the literature and to try to find explanation on the inhibition of calcite growth during the 8h crystallization process, why the size of particles didn’t increase in the 8h aging, the possible explanation of L or D form influence.

Answer:

Thanks for your comments.

When we re-calculated the portion of calcite and vaterite from XRD as you suggested, we found that the result of Figure 2 is not compatible with the re-calculated results.

The reason is that there is a critical limitation in the method used for getting the results of the Figure 2 (in previous version). As mentioned in previous response, we estimated the portion of chiral-curved vaterite by counting the vaterite forms from the low magnification SEM image. That is, we estimated the portion of vaterite by counting the varterite from the over 100 of the generated CaCO3 on the SEM image. I totally agree that this method is not based on the accurate determination of the CaCO3 crystal phases.

I agree and appreciate your suggestion to remove the Figure 2 containing such “inaccurate” results, “for clarity”. We are very sorry that we were inconsiderate and not deeply recognizing the results.

Based on the XRD-based re-calculation, the portion of calcite formation is observed at 8-h incubation. Although the calcite growth might be influenced by the acidic amino acid, the calcite phase can be formed even slowly. 

In the revision, we checked and modified the descriptions about the calcite formation.

I appreciate again your invaluable discussions.

References

Synthesis Methods and Favorable Conditions for Spherical Vaterite Precipitation: A Review, Crystals 2019, 9(4), 223,

Control over the crystal phase, shape, size and aggregation of calcium carbonate via a l-aspartic acid inducing process, Biomaterials 25, (2004), 3923–3929

Reviewer 2 Report

Comments and Suggestions for Authors

The presented research is quite interesting and shows the precipitation of curved vaterite under US treatment where L-enantiomers of Asp induced the production of a curved vaterite structure exhibiting a ‘right-handed’ (counterclockwise) spiraling morphology, whereas D-enantiomers induced the production of structures with a ‘left-handed’ (clockwise). The authors took the comments into consideration and improved the manuscript. The manuscript can be accepted for publication.

Author Response

Thank you for your comments.